# Investigating Metals and Metalloids in Soil at Micrometric Scale Using μ-XRF Spectroscopy—A Case Study

Sofia Barbosa [1,*], António Dias [2], Marta Pacheco [3], Sofia Pessanha [2] and J. António Almeida [1]

1   GeoBioTec GeoBioSciences, GeoTechnologies and GeoEngineering & NOVA FCT (Department of Earth Sciences), Faculdade de Ciências e Tecnologia, 2829-516 Caparica, Portugal
2   LIBPhys & NOVA FCT (Department of Physics), 2829-516 Caparica, Portugal
3   NOVA FCT (Department of Earth Sciences), 2829-516 Caparica, Portugal
*   Correspondence: svtb@fct.unl.pt; Tel.: +351-212-948-573

**Abstract:** Micrometric 2D mapping of distinct elements was performed in distinct soil grain-size fractions of a sample using the micro-X-ray Fluorescence (μ-XRF) technique. The sample was collected in the vicinity of São Domingos, an old mine of massive sulphide minerals located in the Portuguese Iberian Pyrite Belt. As expected, elemental high-grade concentrations of distinct metals and metalloids in the dependence of the existent natural geochemical anomaly were detected. Clustering and k-means statistical analysis were developed considering Red–Green–Blue (RGB) pixel proportions in the produced 2D micrometric image maps, allowing for the identification of elemental spatial distributions at 2D. The results evidence how elemental composition varies significantly at the micrometric scale per grain-size class, and how chemical elements present irregular spatial distributions in the direct dependence of distinct mineral spatial distributions. Due to this fact, elemental composition is more differentiated in coarser grain-size classes, whereas griding-milled fraction does not always represent the average of all partial grain-size fractions. Despite the complexity of the performed analysis, the achieved results evidence the suitability of μ-XRF to characterize natural, heterogeneous, granular soils samples at the micrometric scale, being a very promising investigation technique of high resolution.

**Keywords:** soil matrix; metal distribution per grain fraction; micro-X-ray elemental mapping; RGB clustering image analysis; k-means





## 1. Introduction

Quantification, imaging, and data processing of micro-X-ray Fluorescence (μ-XRF) outputs are presently an interesting but also very challenging area of investigation. To obtain the elemental distribution of a sample, specific instrumentation that provides precise positioning and good energy resolution must be used. Micro-XRF imaging spectrometers rely on scanning samples along the X and Y directions, with a micro-X-ray beam irradiating a region of interest (ROI), point by point [1]. Recent developments in μ-XRF consider quantitative analysis using fundamental parameter-based 'standardless' quantification algorithms [2,3].

The works developed by [2,4–6] evidence the suitability of this technique for various applications within the earth sciences. Further, 2D high-resolution chemical distribution maps can be used as qualitative multi-element maps or as semiquantitative single-element maps through which bulk and phase-specific geochemical data sets can be established [4].

In [2], the authors discuss the accuracy and precision of these quantitative analyses by using a simple-type calibration against a certified reference material of similar matrix and composition. μ-XRF is a non-destructive technique and leaves samples intact for other types of analyses, such as Raman spectroscopy or X-ray diffraction, which allow for the characterization of molecular components [7]. The use of μ-XRF in conjunction with these

established methods of molecular analysis allows for a more complete characterization of grains and particles [2,8,9]. Heterogeneous samples, such as soils, are much harder to characterize. Both single particle as well as bulk analyses must be performed on sample specimens to ensure a full description by μ-XRF [8]. Its consideration to analyse bulk samples of soil implies, necessarily, a clear elemental identification and the distinction between different occurring grades [10]. Quantification of soil data by μ-XRF is still a topic of considerable investigation interest and has been reported only in a limited number of publications [11,12]. Recent research studies evidence how statistical and geostatistical techniques can be applied to co-relate distinct imaging results [13] and how it is already possible to generate 3D maps of chemical properties at the micrometric scale by combining 2D SEM-EDX data with 3D X-ray computed tomography images [14–16]. Effectiveness and potentialities that result from the integration of results of micro-X ray and SEM techniques are also well demonstrated by distinct researchers, even in the cases of very irregular, porous matrixes [14–16]. In fluorescence microscopy, colocalization refers to observation of the spatial overlap between two (or more) different fluorescent labels, each having a separate emission wavelength. Ref. [13] discussed co-localization analysis processes in the context of increasingly popular super-resolution imaging technique occurrence versus correlation, although this limits image pixel-based processing techniques. Ref. [15] developed a method to generate 3D maps of soil chemical properties at the microscale by combining 2D SEM-EDX data with 3D X-ray computed tomography images. The spatial correlation between the X-ray grayscale intensities and the chemical maps made it possible to use a regression-tree model as an initial step to predict 3D chemical composition.

Bulk-sample analysis is a test method used when individual particulate samples are not representative or are not obtained for a certain type of material. Particulate products, such as soils, granulated powders, dusts, or foodstuffs, are usually analysed through bulk-sampling principles [8]. The microscopic analysis of a heterogeneous matrix, such as bulk soil samples, with μ-XRF is complex but has unique potentialities.

The present work is an introductory study in which 2D image clustering analysis based on μ-XRF XY scanning maps of a soil sample was performed. The case study, a soil sample denominated as SD1, was collected at the former mine of São Domingos in Mértola, Portugal (Figure 1). São Domingos Mine is located at the Iberian Pyrite Belt (IPB). It is a world-renowned massive sulphide ore deposit, mainly exploited for its copper contents. High concentrations of As, Zn, and Pb area also found. Its exploitation started prior to the Roman occupation period, mainly for Au, Ag, and Pb. Due to the mine's extensive exploitation over the centuries, the area is filled with very heterogeneous mining waste. Natural gossan (iron caps) deposits and natural local mineralogy results in the generation of heterogeneous soils with high contents of several heavy metals and metalloids. At this mining site, the geology is dominated by greywackes and quartzwackes, quartzites, phyllites, schists, forming the "Baixo Alentejo" Flysch Group, turbidites, and a volcano–sedimentary complex. The lithostratigraphic units range mainly from the Devonian to the Carboniferous periods [17,18]. Due to its mining context and its local geology, the most common elements found in the soils around the mining area are, mainly, Al, Si, S, Ti, Mn, Cr, Fe, Cu, Zn, As, Ga, Pb, Sb, and Hg [19,20].

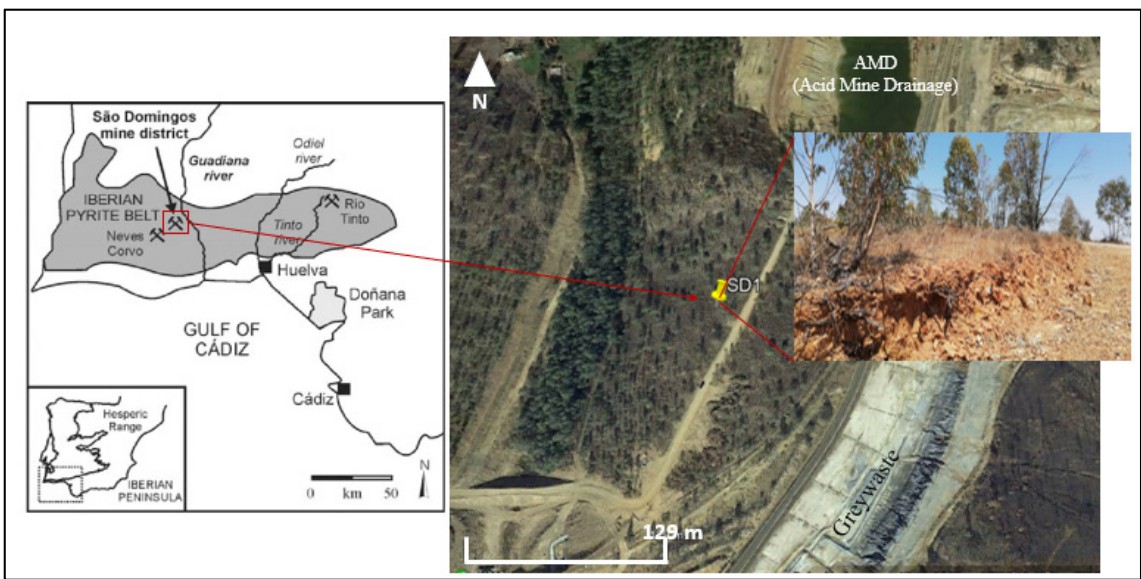

**Figure 1.** Location of São Domingos mine and location of the collected SD1 sample. Left figure is adapted from [17].

## 2. Materials and Methods

### 2.1. Sampling and Sample Preparation

The soil sample was collected with the aid of a small shovel, scooping the surface soil to a depth of about 10 to 20 cm. About 1.50 kg of material was collected, stored, and labelled adequately. SD1 consists of a reddish-brown soil with small to large particles (Figure 2). The sample was sieved into four classes of grain size, ≥2 to <3 mm, <2 mm to ≥500 µm, <500 µm to ≥250 µm, and <250 µm. A ground and milled bulk sample (TM, "Total Milled") was also prepared. Depending on the availability of the material, and using a manual benchtop press, two to five pellets were made from all the granulometry-size fractions and TM. Table 1 shows the number of pellets analysed by category. These pellets were analysed with a benchtop micro-XRF spectrometer, M4 TORNADO by Bruker (Billerica, MA, USA).

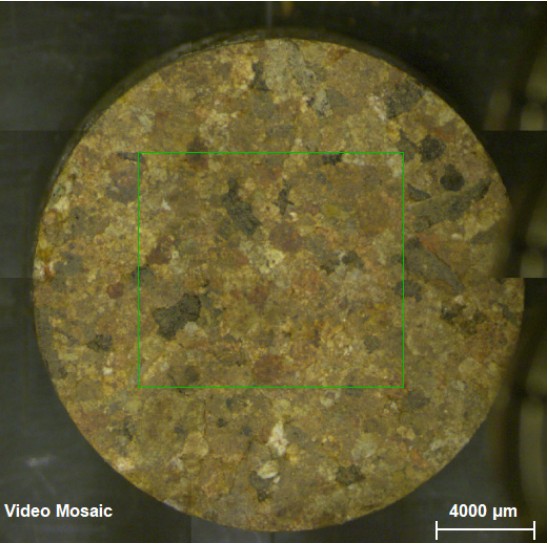

**Figure 2.** Pellet of an original SD1 sample of grain size fraction "<2 mm to ≥500 µm" (image source: Bruker's M4 TORNADO camera).

**Table 1.** Number of pellets by category.

| Categories/Fraction | Number of Pellets (SD1) |
| --- | --- |
| TM | 2 |
| $\geq$2 mm to <3 mm | 2 |
| <2 mm to $\geq$500 μm | 5 |
| <500 μm to $\geq$250 μm | 3 |
| <250 μm | 3 |

### 2.2. Micro X-ray Fluorescence Multi-Point Measurements and 2D Image Mapping

The micro-X-ray fluorescence technique is applied by means of the energy dispersive spectrometer M4 TORNADO by Bruker. This instrument consists of a low-power X-ray tube with a Rh anode, which was operated in this case study at 50 kV and 300 uA. Placed after the X-ray tube, a poly-capillary lens focuses the beam to a spot size that can go down to 25 μm for Mo-K$\alpha$. This way, by selecting an area in the sample, point-by-point measurements can be performed and images of elemental distributions within the sample are generated.

In the case study, the pellets were analysed making use of an AlTiCu 100/50/25 μm filter composition. For elements emitting radiations from 5 to 35 keV, it is adequate to use filters that can lessen the effect of the Bremsstrahlung radiation that contribute to background radiation [21]. Therefore, for SD1, the two filters mentioned above were used due to the presence of elements with an atomic number (Z) superior to 21, i.e., from Titanium (Ti) to Yttrium (Y), which were identified in a primary analysis without filters.

The measurements were taken under 20 mbar vacuum conditions (to improve detection limits), with a step size of 15 μm and 10 ms acquisition per spectrum rendering for, on average, 1 h 30 min to ensure high-resolution 2D maps for each element.

Data treatment of micro-2D mapping was performed using the M4 TORNADO inbuilt software MQuant.

That is to say, only one pellet for each of the sample categories—TM, $\geq$2 mm to <3 mm, <2 mm to $\geq$500 μm, <500 μm to $\geq$250 μm, <250 μm—was chosen for 2D map surveys due to the big amount of data obtained.

### 2.3. Two-Dimensional Image Mapping Processing: Clustering RGB Pixel Analysis

μ-XRF 2D mapping outputs consisted of 2D image files. Possibilities related with the processing of these image files are mainly related with pixel quantification and statistical analysis of its distributions. In this case study, each image refers to a certain element spatial distribution for which its occurrence and concentration are locally represented by a certain intensity of a certain RGB (Red, Green, Blue) colour. The highest elemental concentrations are represented by the highest RGB light colour proportions (Figure 3).

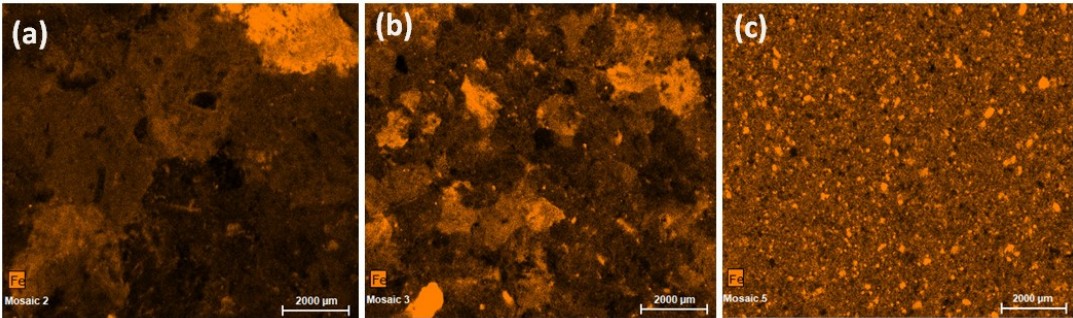

**Figure 3.** μ-XRF 2D mapping outputs for the element Iron (Fe). (**a**) Grain-size distribution: $\geq$2 mm to <3 mm; (**b**) <2 mm to $\geq$500 μm; (**c**) <250 μm.

Pixel proportion quantifications per distinct RGB colour intensity were established with R©Countcolors Package [22–25]). This package was developed originally with the aim

of quantifying the area of white-nose syndrome infection of bat wings [25]. R©Countcolors Package allows users to quantify regions of an image by distinct colours. It is an R package that counts colours within specified colour ranges in image files and provides a masked version of the image with targeted pixels changed to a different selected colour by the utilizer. This package integrates techniques from image processing without using any machine learning, adaptive thresholding, or object-based detection, which make it reliable and easy to use but limited in terms of application.

The principle of the image processing analysis consisted of considering each RGB colour in three dimensions, where each colour is defined by its coordinates in R (red), G (green), and B (blue) axes. The range of each RGB colour is, thus, interpreted in a 3D space (Figure 4a). The quantitative RGB pixel analysis performed for each 2D image begins with the verification of the level of similarity of colour intensities according to its respective RGB code. Each RGB code represents a certain RGB cluster (and, thus, a certain colour intensity). RGB pixels per cluster are counted by samples of 10,000 pixels from the 2D image. For each RGB code representing a certain cluster, its respective frequencies are calculated (Figure 4b,c). Figure 4 presents an exemplification of the pixel-counting frequencies for six distinct colour clusters representing the concentrations of the element Fe. The pixels of more light-colour clusters represent the locations with highest concentrations on Fe. The number of clusters and the number of the sampling pixels are established by the user.

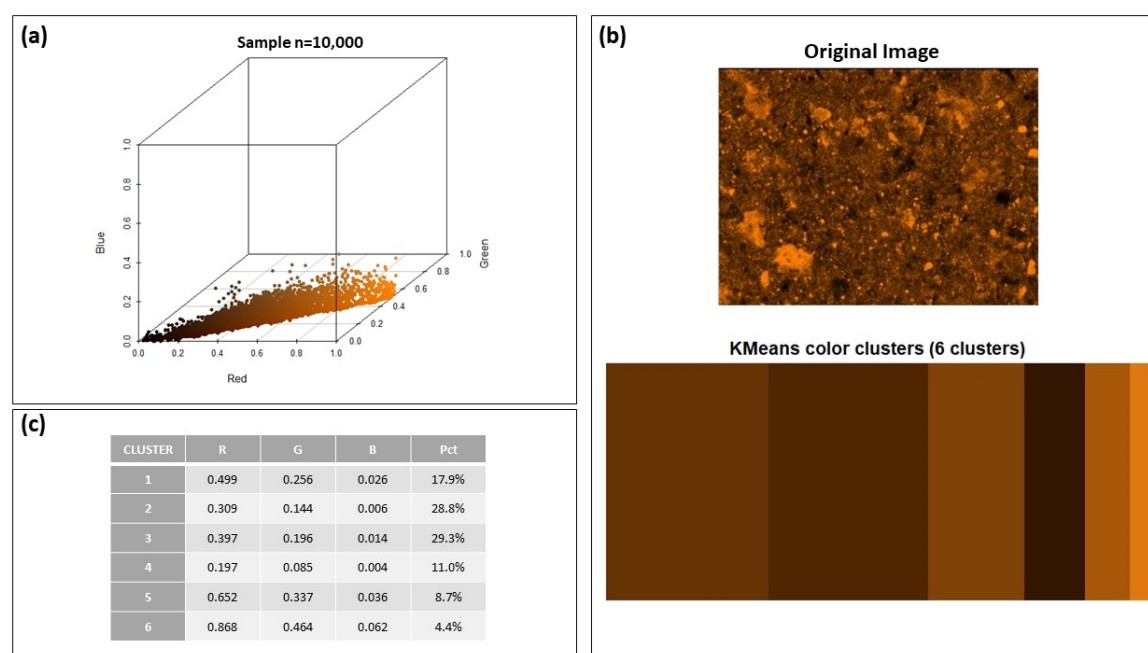

**Figure 4.** RGB clustering analysis of a μ-XRF 2D map (element: Fe; pellet of a bulk sample). (**a**) RGB counting colours in three dimensions (sample size n = 10,000 pixels); (**b**) Pixel classification in 6 clusters; (**c**) RGB pixel proportions for each cluster.

One of the main objectives of this case study was to estimate the areas that are associated with a certain range of RGB pixels. The light-colour ranges that are associated with the highest colour intensities represent the highest elemental concentrations. In the adopted methodology, after selecting the colour clusters that are the most representative for a certain element occurrence, its respective areas are estimated. The images processed always integrate degrees of intensity of a unique colour, which relates to a certain element to be identified. The element occurrence is represented by the light-coloured clusters in each colour image. In [5], following the principles described in [23–25], the authors defined an analysis methodology based on two options: one that considers upper and lower limits for each colour range and where a box-shaped border is drawn around the region of that range

(rectangular range) and a second option that considers the selection of a certain central colour and a search radius around it, were a "sphere" for the considered colour range is drawn (spherical range). Due to the possibilities of applying distinct criteria, estimated area calculations are referenced in terms of percentages of minimum and maximum probable areas (Figure 5). In fact, the calculated areas have distinct possibilities, directly dependent on the number of colour clusters and the search criteria, which are, in turn, user defined. Due to these distinct possibilities, it is more correct to suggest a range of probable estimated areas than to present only a specific estimated area. For this, the adopted methodology integrates the possibility of considering the search criteria to one, two, or three colour clusters simultaneously (Figure 5). When two or three colour clusters are to be considered, a search radius is applied to each colour. For minimum area calculations, it is advisable to consider "one colour cluster" with spherical or rectangular search criteria or "two colour clusters" procedures. To calculate possible maximum estimated areas, it is advisable to simultaneously consider "three colour clusters" for the estimations.

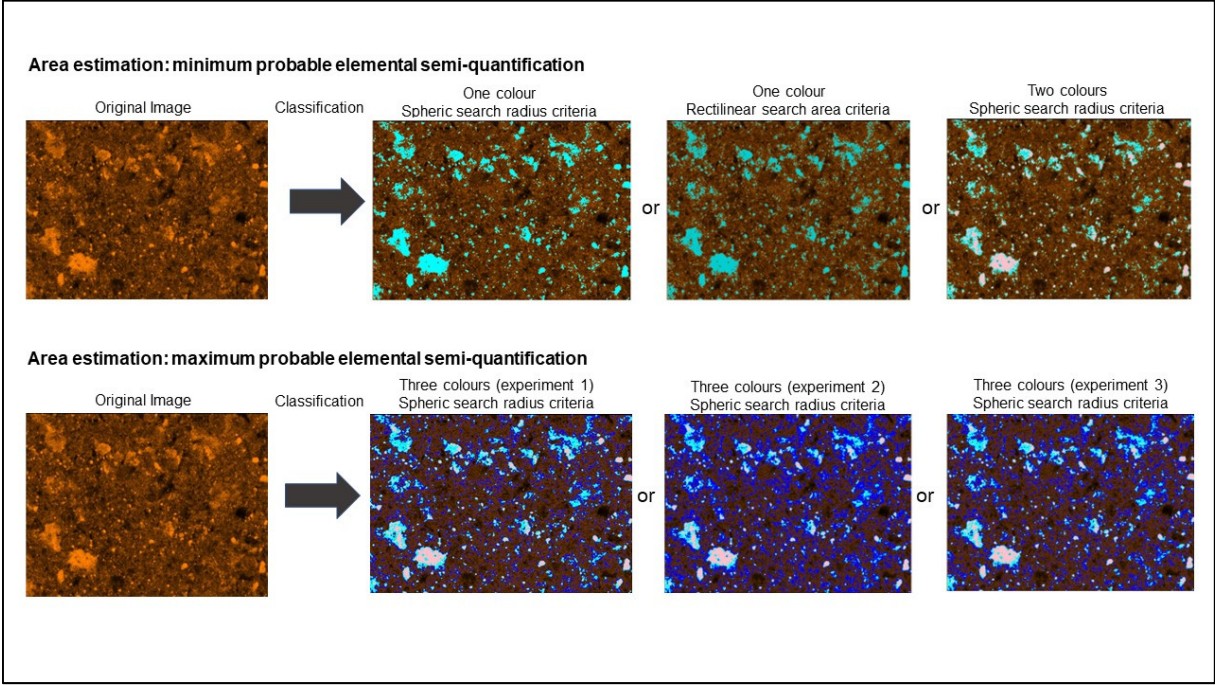

**Figure 5.** Methodology applied to estimate minimum and maximum probable elemental occurrence in a μ-XRF 2D map (example of element Fe in a bulk pellet sample).

This methodology allows one to accomplish a semi-quantitative analysis of the μ-XRF 2D mapping images. Uncertainty is mostly associated with the clustering classification and search criteria, which are user defined. The described methodology has already been applied to granular mining waste samples [5] and to a syenite nepheline rock sample in order to identify incompatible and scarce metals at the micrometric scale [5]. Results evidence the potentiality of this methodology to interpret elemental μ-XRF 2D mapping images of materials with heterogenous granular textures, such as soils and mining wastes, being also quite promising in elemental and mineral identification of distinct rock matrix [5,6].

### 3. Results—Elemental μ-2D Mapping Distributions

Through multi-point measurement analysis, it was possible to identify, in sample SD1, the following elements per size fraction class: aluminium (Al), silicon (Si), potassium (K), calcium (Ca), titanium (Ti), manganese (Mn), iron (Fe), nickel (Ni), copper (Cu), zinc (Zn), gallium (Ga), arsenic (As), rubidium (Rb), strontium (Sr), and yttrium (Y). Figure 6 presents the results achieved for the methodology applied for the case of the element Fe.

Estimations of minimum and maximum probable Fe occurrence in μ-XRF 2D maps are presented. Analogous results are presented for the elements Ca, Mn, Cu, Zn, and As in Appendix A.

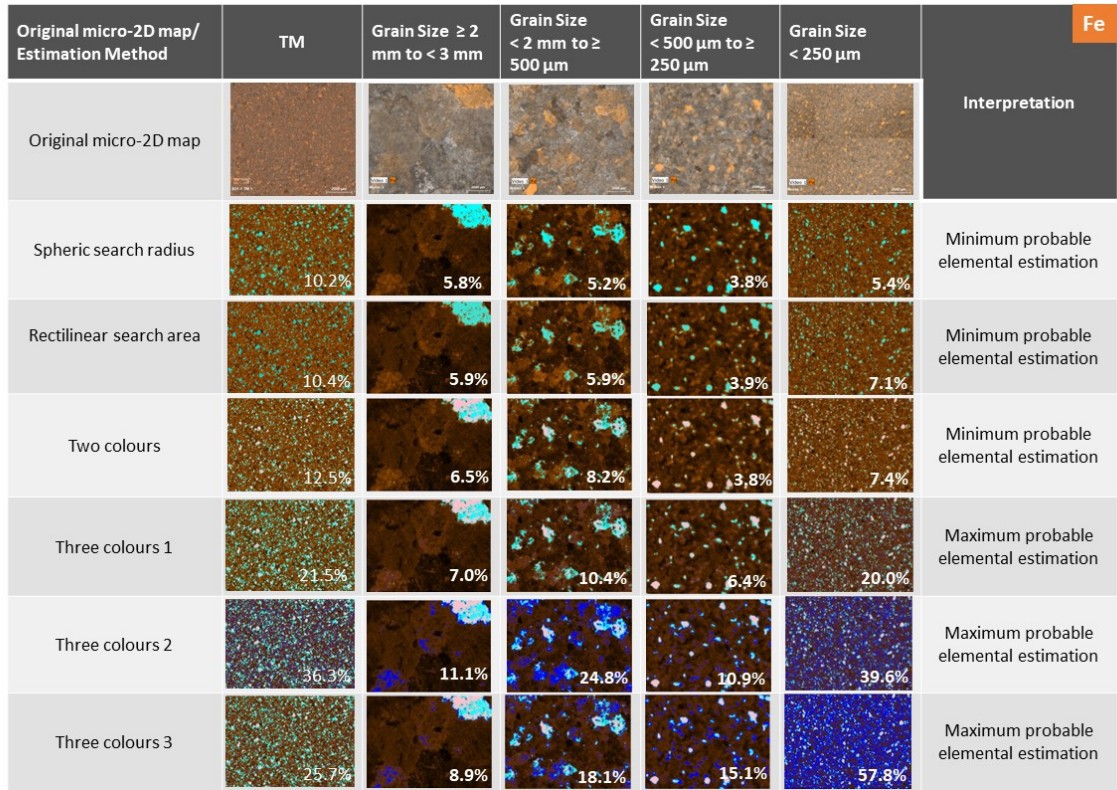

**Figure 6.** Minimum and maximum probable elemental occurrence in μ-XRF 2D map (percentage of area %) for Fe.

As can be observed, the difference in spatial distribution patterns and the estimated minimum and maximum elemental quantities is clear according to grain-size fractions. Further, patterns of TM (ground and milled) are more similar to grain-size fraction "<250 μm". This behavioural pattern can be observed in most of the analysed elements (Appendix A). Quantities per element are estimated in percentage (%) of the total mapped area and vary according to grinding, milling, and grain-size fraction (Figure 6, Appendix A and Figure 7). Bulk milled samples do not always represent the average between the distinct size fractions. In fact, for some elements, coarser gain-size fractions, such as "≥2 mm to <3 mm" and "<2 mm to ≥500 μm", tend to be present in distinct estimated quantities (Figure 6, Appendix A and Figure 7). These two facts are indicative of the occurrence of some elements in the direct dependence of the mineralogy and, in turn, in the dependence of its more representative granulometry. Table 2 includes a summary of the minimum and maximum elemental occurrence in the μ-XRF 2D map (percentage of area, %) of the elements Al, Si, K, Ca, Ti, Mn, Fe, Ni, Cu, Zn, Ga, As, Sr, and Y.

Elements presented in higher estimated percentages evidence the influence of the local geology in the soil's constitution [17,26–28]. Figure 8 presents some of the most representative results considering maximum estimated percentages of elemental occurrence area (%). The elements presented in this Figure, Si, Al, Cu, Zn, Ca, K, Ti, Fe, As, Ga, and Mn, are grouped according to their respective percentage of occurrence area (%). The results reflect not only the natural composition of soil (Si, Al, Ca, K, Ti) but also the presence of natural geochemical anomalies, which are related to the existence of massive sulphide ore deposit minerals, increasing the percentages of occurrence of Cu, Zn, Fe, As, Ga, and Mn among other elements. Apart from Si and Al, the elements Cu, Zn, Ca, K, Ti, Fe, As,

Ga, and Mn present specific spatial distribution patterns. For the case of Fe, As, Ga, and Mn, the dependence on coarser minerals is quite evident. Spatial overlap of the elements according to mineralogy is also possible to observe. In this context, the spatial overlap of Fe, As, and Ga is an example and is a consequence of the local geochemistry and mineralogy, which includes iron oxides and sulphides [17,26–28]. Simultaneously, the presence of As and Fe can be explained by the existence of arsenic-bearing sulfides, such as arsenopyrite or sulfosalts. The presence of Ga in the soil is usually connected with the occurrence of silty minerals. Ga tends to be sorbed by Fe(III) and Mn(III) oxides [29,30] and occurs as an impurity in iron oxides, hydroxides, and sphalerite minerals, which can explain the spatial correspondence between Fe and Ga in the SD1 sample.

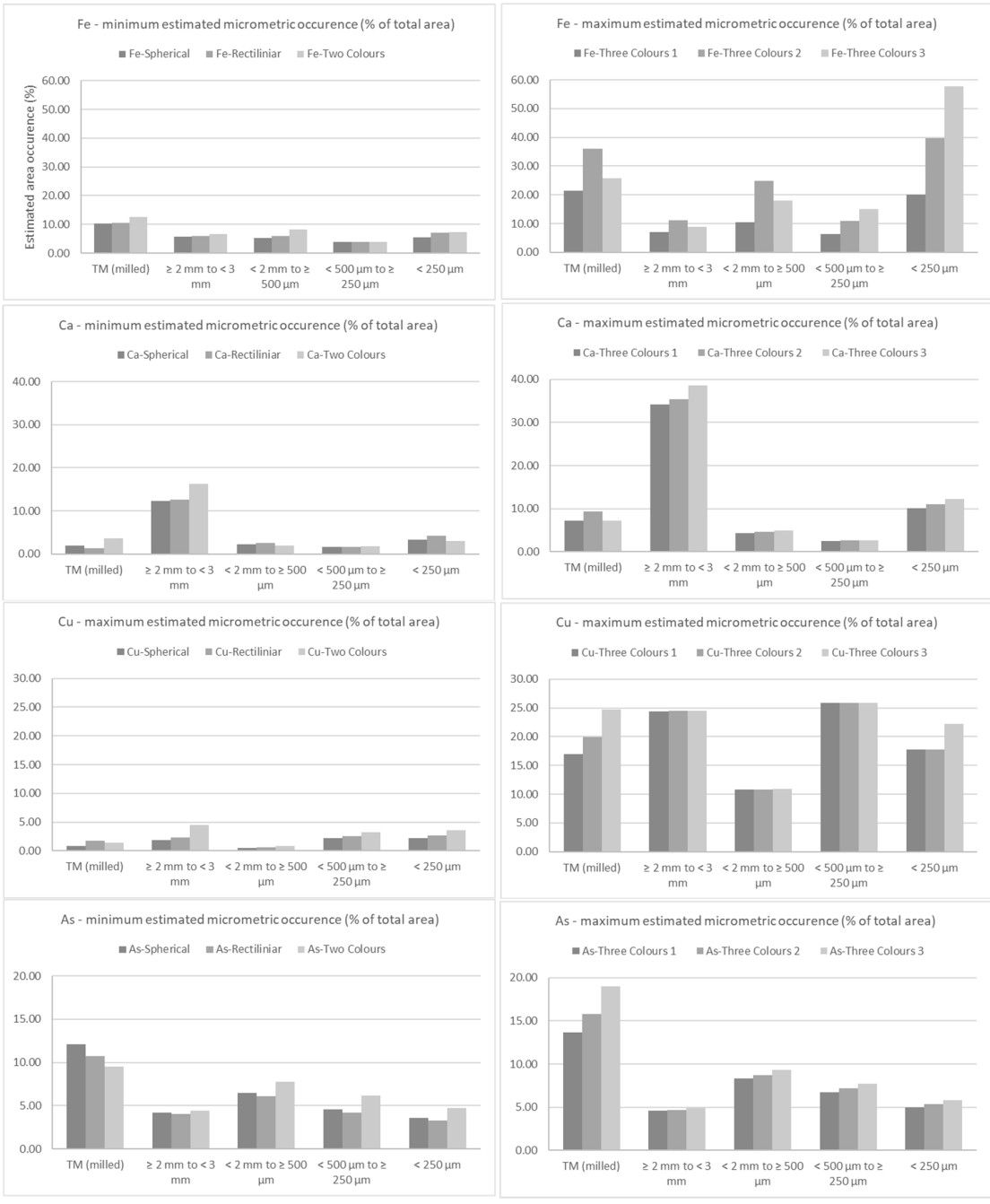

**Figure 7.** Minimum and maximum elemental occurrence in μ-XRF 2D map (percentage of area %) for Fe, Ca, Cu, and As.

**Table 2.** Synthesis of estimated minimum and maximum elemental occurrence (percentage of area %) for Al, Si, K, Ca, Ti, Mn, Fe, Ni, Cu, Zn, Ga, As, Sr, and Y.

| Sample Fraction | TM (ground and milled) | | Grain Size ≥2 mm to <3 mm | | Grain Size <250 μm | |
|---|---|---|---|---|---|---|
| Element | Minimum probable | Maximum probable | Minimum probable | Maximum probable | Minimum probable | Maximum probable |
| Al | 6.6–7.8 | 18.3–27.8 | 8.5–9.8 | 20.3–29.7 | 11.5–13.2 | 26.4–37.0 |
| Si | 3.8–4.4 | 5.4–5.5 | 14.8–19.1 | 24.5–24.9 | 26.1–32.1 | 41.6–42.2 |
| K | 5.8–7.8 | 35.8–44.7 | 2.3–2.8 | 10.4–13.3 | 5.4–8.8 | 35.9–44.2 |
| Ca | 2.0–3.6 | 7.3–9.4 | 12.3–16.3 | 34.1–38.6 | 3.1–4.3 | 10.1–12.3 |
| Ti | 2.1–4.6 | 3.3–3.5 | 7.3–9.6 | 10.0–14.0 | 5.9–6.2 | 5.6–6.9 |
| Mn | 2.4–12.7 | 19.8–21.5 | 0.9–1.4 | 1.7–1.8 | 1.9–9.3 | 13.5–14.8 |
| Fe | 10.2–12.5 | 21.5–36.3 | 5.8–6.5 | 7.0–11.1 | 5.4–7.4 | 20.0–57.8 |
| Ni | 3.2–7.5 | 15.3–24.4 | 3.8–8.8 | 17.7–27.9 | 9.3–16.0 | 28.0–39.3 |
| Cu | 0.8–1.7 | 17.0–24.7 | 1.9–4.5 | 24.4–24.5 | 2.2–3.6 | 13.5–14.8 |
| Zn | 11.2–12.9 | 49.4–59.7 | 11.0–19.6 | 30.9–37.4 | 7.0–18.1 | 33.2–41.9 |
| Ga | 4.8–9.4 | 10.6–16.2 | 3.0–4.4 | 4.7–6.8 | 7.0–15.7 | 17.4–24.2 |
| As | 9.5–12.1 | 13.7–19.0 | 4.0–4.4 | 4.6–4.9 | 3.3–4.7 | 5.0–5.8 |
| Sr | 13.8–19.6 | 42.8–49.5 | 22.2–27.3 | 41.9–45.4 | 28.0–36.1 | 54.0–54.1 |
| Y | 3.4–4.4 | 5.6–7.3 | 7.6–9.4 | 11.1–13.3 | 7.8–9.6 | 11.6–13.9 |

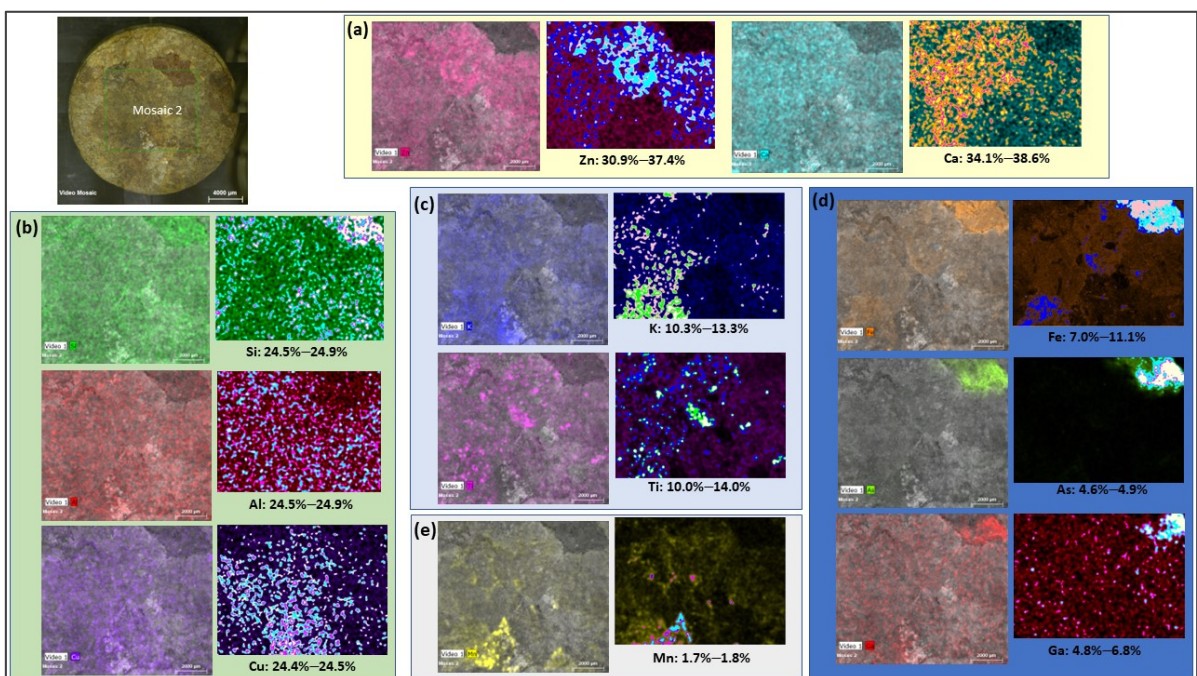

**Figure 8.** Image 2D micrometric maps of the elements (**a**) Zn, Ca (**b**) Si, Al, Cu (**c**) K, Ti (**d**) Fe, As, Ga (**e**) Mn in sample SD1, grain-size fraction "≥2 mm to <3 mm", and correspondent maximum estimated percentages of occurrence area (%).

## 4. Discussion and Conclusions

Elemental 2D spatial mapping through micro-XRF spectroscopy is a promising technique in the detailed study of granular heterogeneous samples, such as soils and mining wastes [31–33]. In this exploratory study, a clustering image analysis methodology was applied to detect elemental distribution at micrometric scale according to distinct colour intensities. The results present accurate information on the elemental distribution per grain fraction, offering clues of its geochemical occurrence (manly primary in coarse grain-size fraction and secondary in finer fractions). Results are more regular and similar between distinct fraction samples and milled samples when the element occurs at lower granulome-

tries. The results showed that the elemental spatial patterns per grain-size fraction are not always coincident or similar to grinding and milled spatial pattern samples, showing that, for some cases, elemental distribution is dependent on specific mineralogy, which can have its own grain-size distribution pattern according to geochemical characteristics of the site. Some metals show distinctive percentages of occurrence according to grain-size fraction. Metal occurrence in milled fractions do not always correspond to the average of the grain-size fractions. Certain elements tend to be present in higher quantities in coarse fractions, mainly 2–3 mm, while other elements tend to present in smaller-size grain fractions (<250 μm). This will be dependent on the mineralogy and specific geochemical behaviour, especially mobility, of the elements. For sure, mobility and geochemical source of the element (primary or secondary) will dictate elemental specific spatial patterns at the micrometric scale.

In general, minimum and maximum elemental estimations from 2D maps show a tendency of greater discrepancies in results when the element is more abundant and widespread in the matrix. This is the example of element Si, K, Zn, Sr, and finer gain-size fractions of Fe. Major discrepancies in measurements are due to the higher difficulty in fixing the characteristic degree of colour intensity that marks the occurrence of the element, and distance between the distinct intensity colour degrees, which may make clustering classification difficult. In this context, the joint interpretation of 2D images to estimate 3D grades is currently an emerging research area [13,14,31–33] that will represent a quite interesting investigation upgrade.

The exploration of applicable data image analysis techniques able to identify elemental spatial overlaps in μ-XRF 2D map surveys and the estimation of grain-size distributions per element or per groups of elements are two promising areas for forward investigation in granular and heterogeneous samples, such as in the case of soil samples.

**Author Contributions:** Conceptualization, S.B. and A.D.; methodology, S.B. and A.D.; software, S.B. and M.P.; validation, S.B., A.D., S.P. and J.A.A.; investigation, S.B., A.D. and M.P.; writing—original draft preparation, S.B.; writing—review and editing, A.D., S.P. and J.A.A. All authors have read and agreed to the published version of the manuscript.

**Funding:** This research was funded by FCT-Fundação para a Ciência e a Tecnologia, Portugal, grants number UIDB/04035/2020, and UID/FIS/04559/2020.

**Institutional Review Board Statement:** Not applicable.

**Informed Consent Statement:** Not applicable.

**Data Availability Statement:** Not applicable.

**Acknowledgments:** The authors acknowledge the support of LIBPhys, GeoBiotec, Department of Physics and Department of Earth Sciences of Nova School of Science and Technology for the development of the laboratory work.

**Conflicts of Interest:** The authors declare no conflict of interest.

## Appendix A

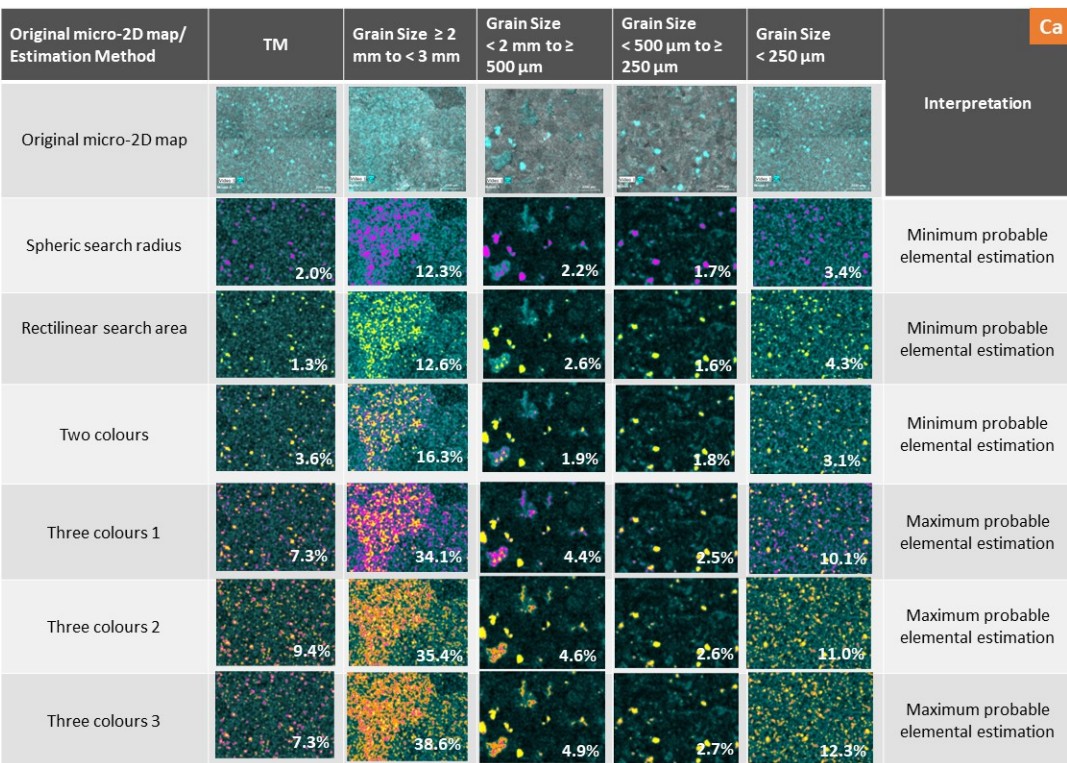

**Figure A1.** Minimum and maximum probable elemental occurrence in µ-XRF 2D map (percentage of area %) for Ca.

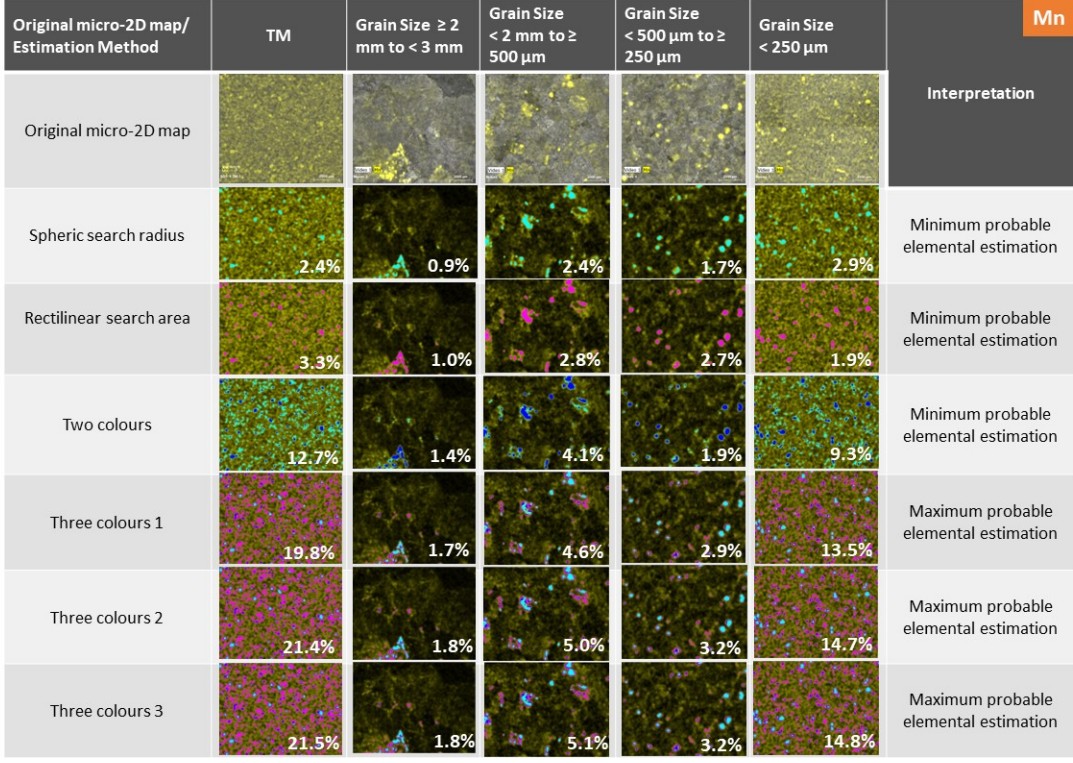

**Figure A2.** Minimum and maximum probable elemental occurrence in µ-XRF 2D map (percentage of area %) for Mn.

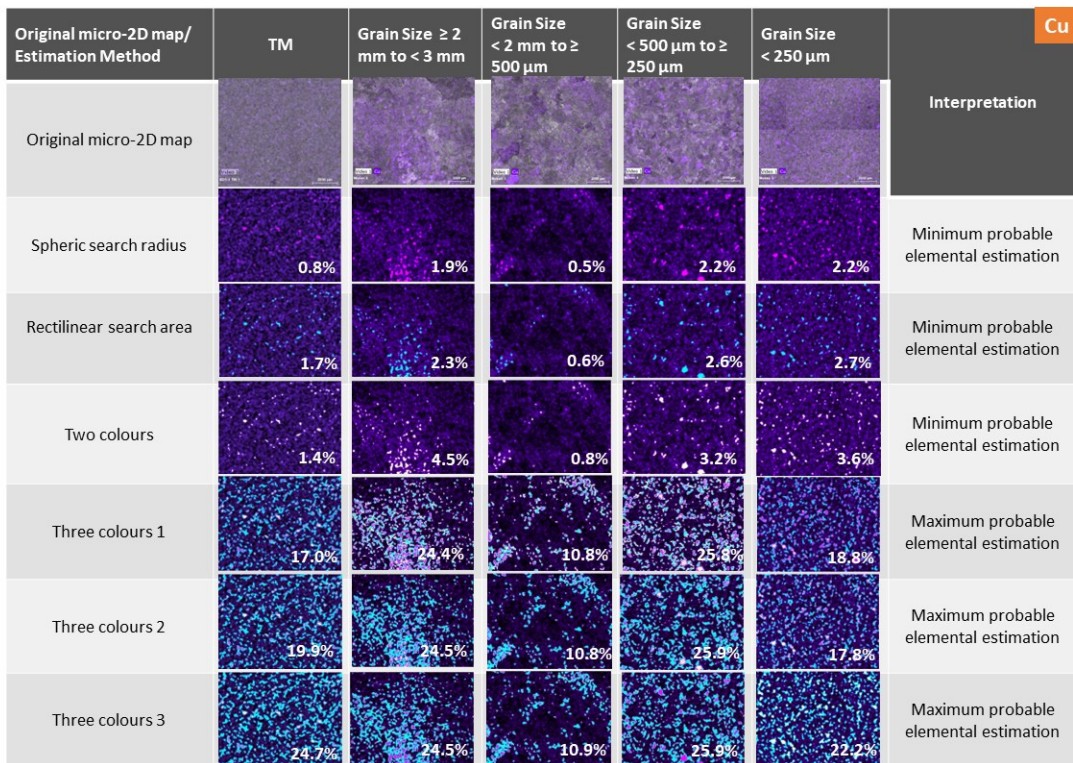

**Figure A3.** Minimum and maximum probable elemental occurrence in μ-XRF 2D map (percentage of area %) for Cu.

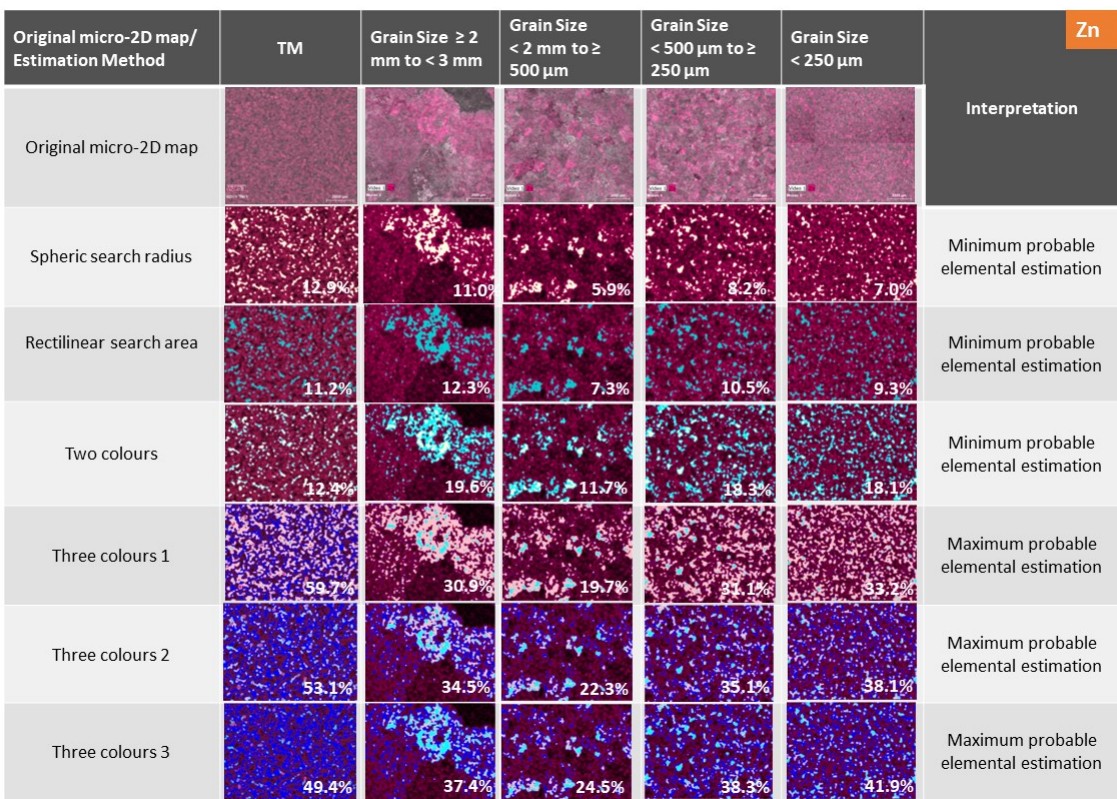

**Figure A4.** Minimum and maximum probable elemental occurrence in μ-XRF 2D map (percentage of area %) for Zn.

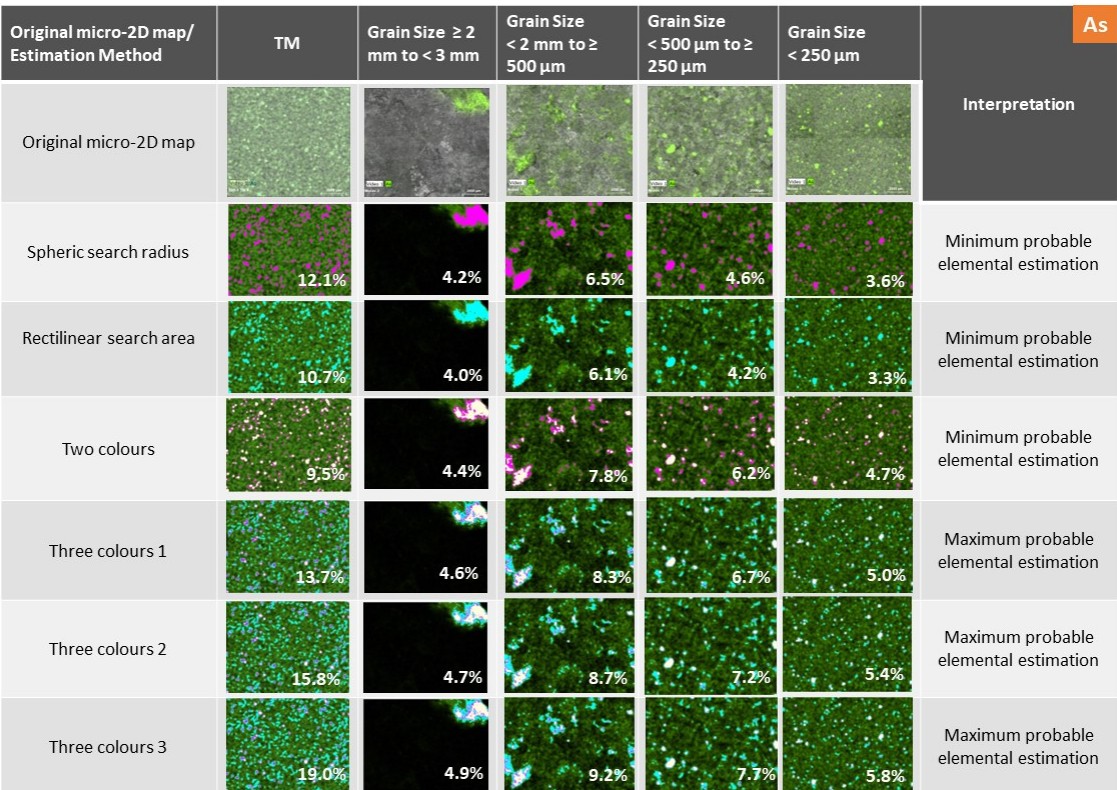

**Figure A5.** Minimum and maximum probable elemental occurrence in μ-XRF 2D map (percentage of area %) for As.

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
