# Peer review of "Investigating Metals and Metalloids in Soil at Micrometric Scale Using µ-XRF Spectroscopy—A Case Study"

_2673-4117, doi:10.3390/eng4010008_

Round 1

Reviewer 1 Report

Article has numerous grammatical errors and typos.  Please work on correcting the English grammatical issues. 

Line 96: check the number value with the kg units

Line 141 and 142: the individuals named here, are they part of a company or university?  If so, please add additional information

Table 2: not sure if this is an issue in the pre-print only but make sure this table fits on 1 page.  Hard to follow without the headings for most of the elements presented.

Author Response

Dear Editors,

Dear Reviewers,

Thank you very much for your consideration, contributions, and comments. The manuscript has been reviewed following your comments, corrections, and suggestions. As suggested by some of the reviewers, an extensive editing of English language and style was performed to our manuscript.

We hope that these revisions are accordingly with your expectations, and we thank you for your detailed revision work which has significantly improved the quality of our manuscript.

Thank you in advance,

Sofia Barbosa, corresponding author

Detailed response list in the attached PDF file.

Reviewer 2 Report

Using the micro-X Ray Fluorescence technique, the present work investigates, at the micrometric scale, metals and metalloids in the soil sample collected near Sao Domingos. Authors have developed a statistical analysis method to identify and correlate the elemental 2D spatial distributions. The obtained 2D mapping reveals that the elemental composition varies significantly at the micrometric scale for any grain size class, and the elements have irregular spatial distributions.

The experiments and the proposed analysis method presented in this paper is useful to characterize natural, heterogeneous, granular soil samples at the micrometric scale. The article has too many long sentences, and there are also many grammatical errors. Considering the large amount and valuable content of the article, it is worth publishing after careful revision. In the following, some errors are listed.

Page 1, line 13, change “were” to “was”

Page 1, line 16, it should be “As expected, …”

Page 1, line 20, add “the” before “micrometric scale…”

Page 1, line 23, it should be “… fraction does not always represent the average…”

Page 1, line 25, it should be “… granular soil samples at the micrometric scale …”

Page 4, line 110, it should be “… consists of a low …”

Page 4, line 114, please change to “… the sample is generated.”

Page 11, line 275, “element” should change to “elements”

Author Response

(The authors gave the same response as above.)

Reviewer 3 Report

The papaer by Barbosa et al., presents an interesting tool for analysis of 2D elemental maps.

The case study is interesting but the descriptoin of the analysis itself could be improved.

In Fig. 4a it looks like very few pixels are way higher (along the blue coordinate axis) than all the others,
I wonder what happens among the vast majority of the pixels: are there any differences in height in those? If so, mabe a different scaling would help in showing it. If not, why? 

A careful spell-check should be made.
A few typos that must be corrected include: line 37 (considers), line 44 (evidence), line 114 (in generated), line 175 (were), 201 (potash), 209 (maximal), 216 (tends), 217 (these), 229 (represents), 230 (evidence), 234 (reflects), 256 (analyses)

Percentage of the third row of fig.6 has to be fixed: 7.1% - which I suppose belongs to the 3rd row - is reported in the 4th row.

Author Response

(The authors gave the same response as above.)
